# Three-Phase Mixed Titania Powder Modified by Silver and Silver Chloride with Enhanced Photocatalytic Activity under UV–Visible Light

**DOI:** 10.3390/nano12091599

**Published:** 2022-05-09

**Authors:** Xiaodong Zhu, Fengqiu Qin, Yangwen Xia, Daixiong Yang, Wei Feng, Yu Jiao

**Affiliations:** 1School of Mechanical Engineering, Chengdu University, Chengdu 610106, China; xiaodangjia21@126.com (X.Z.); mysumeiren@163.com (F.Q.); x1278704108@163.com (Y.X.); yangdaixiong1998@163.com (D.Y.); 2School of Science, Xichang University, Xichang 615000, China

**Keywords:** TiO_2_, three-phase mixed structure, Ag/AgCl modified, hydrothermal method, photocatalytic performance

## Abstract

Pure and Ag/AgCl-modified titania powders with anatase/rutile/brookite three-phase mixed structure were prepared by one-step hydrothermal method. The effects of Ag/Ti atomic percentages on the structure and photocatalytic performance of TiO_2_ were investigated. The results showed that pure TiO_2_ consisted of three phases, anatase, rutile, and brookite, and that Ag addition promoted the transformation from anatase to rutile. When the molar ratio of Ag/Ti reached 4%, the AgCl phase appeared. The addition of Ag had little effect on the optical absorption of TiO_2_; however, it did favor the separation of photogenerated electrons and holes. The results of photocatalytic experiments showed that after Ag addition, the degradation degree of rhodamine B (RhB) was enhanced. When the molar ratio of Ag/Ti was 4%, Ag/AgCl-modified TiO_2_ exhibited the highest activity, and the first-order reaction rate constant was 1.67 times higher than that of pure TiO_2_.

## 1. Introduction

Because of the relatively low utilization of sunlight and photogenerated charge separation rate of pure TiO_2_, the modification of TiO_2_ to improve its photocatalytic performance is a research hotspot [1,2,3,4,5]. Reports have shown that ion doping [1,6,7,8], semiconductor combination [4,9,10], noble metal modification [11,12,13], and other modification methods can effectively enhance the photocatalytic properties of TiO_2_. Precious metal particles deposit on the surface of TiO_2_ to form heterojunctions, which show a surface plasmon resonance (SPR) effect due to the resonance between the electron on the surface of the precious metal and the light wave under illumination, enhancing the visible region absorption and improving the sunlight utilization [11,12]. On the other hand, as the Fermi level of noble metal is lower than the position of the TiO_2_ conduction band, the electrons in the TiO_2_ conduction band migrate to the surface of noble metal particles, reducing the probability of recombination with the valence band holes, which contributes to improvement in quantum efficiency [13].

Among precious metals, Ag-modified TiO_2_ is a focus of research [2,14,15,16,17,18]. In recent years, on the basis of Ag modification, Ag/AgX (X = Cl, Br, I)-modified TiO_2_ has been developed. On one hand, the SPR effect of Ag can be used to improve the utilization of visible light. On the other hand, the presence of AgX is conducive to the separation of photogenerated charges [19,20,21,22]. Jing et al. [20] prepared Ag/AgI-modified TiO_2_ by a solvothermal method and found that the activity of Ag/AgI@TiO_2_ was higher than that of AgI@TiO_2_ and pure TiO_2_, regardless of whether it was irradiated by ultraviolet or visible light. Yu et al. [21] synthesized Ag/AgCl@TiO_2_ by combining a precipitation method and a photoreduction method. After Ag/AgCl modification, the visible light absorption was increased and the recombination of photogenerated electrons and holes was significantly reduced. Thus, the photocatalytic performance was improved.

Generally, TiO_2_ shows three crystal structures: anatase, rutile, and brookite. When they combine, a mixed-crystal effect results, and better photocatalytic activity than with a single crystal can be achieved [23,24]. Anatase/rutile mixed crystals have been studied widely [10,25,26,27]. On the basis of two-phase mixed crystals, anatase/rutile/brookite three-phase mixed structures can be formed, which can further enhance the photocatalytic activity [28,29,30]. In the present study, a one-step hydrothermal method was used to synthesize Ag/AgCl-modified anatase/rutile/brookite three-phase mixed crystal photocatalytic materials. The effects of the Ag/Ti molar ratio on the crystal structure, morphology, element chemical state, surface area, bonding, optical properties, and photocatalytic performance of the obtained photocatalysts powders under UV–visible light were studied.

## 2. Experiment

### 2.1. Material Preparation

Polyethylene glycol (Analytical Reagent, AR), butyl titanate (AR), anhydrous ethanol (AR), hydrochloric acid (AR), silver nitrate (AR), and rhodamine B (AR) were purchased from Chengdu Chron Chemicals Co., Ltd. (Chengdu city, Sichuan province, China). All the chemical reagents were used directly without further purification.

First, 10 mL butyl titanate was dissolved into 30 mL anhydrous ethanol to obtain solution A. Then, 1 mL hydrochloric acid and 1 mL polyethylene glycol were dissolved into 30 mL deionized water to acquire solution B, which was added dropwise into solution A. After continuous stirring, the mixed solution was transferred to the hydrothermal reaction kettle for hydrothermal treatment at 190 °C for 20 h. After precipitation and washing, the pure TiO_2_ powder photocatalyst was obtained by drying in the oven at 80 °C.

Quantities of 0.0482 g, 0.0964 g, 0.1928 g and 0.2892 g AgNO_3_ were added to solution B, and the other preparation steps were the same to gain Ag- and Ag/AgCl-modified TiO_2_. The Ag/Ti atomic percentages were 1%, 2%, 4%, and 6%, respectively. The modified TiO_2_ were marked as x%Ag-TiO_2_ (x = 1, 2, 4, and 6).

### 2.2. Characterization Techniques

A DX-2700 X-ray diffractometer (XRD, Dandong Haoyuan Instrument Co. Ltd., Dandong, China) was used to characterize the crystal structure of the sample. A Hitachi SU8220 scanning electron microscope (SEM) and JEM-F200 transmission electron microscope (TEM and HRTEM) were employed to observe the surface morphology (FEI Company, Hillsboro, OR, USA). An XSAM800 multifunctional surface analysis system (XPS, Thermo Scientific K-Alpha, Kratos Ltd., Manchester, UK) was used to analyze the composition and valence of elements. A Mike ASAP2460 analyzer (BET, Mike Instrument Company, Atlanta, GA, USA) was used to measure the specific surface area. An F-4600 fluorescence spectrum analyzer with an excitation wavelength of 320 nm (PL, Shimadzu Group Company, Kyoto, Japan) was used to detect the recombination of photogenerated electrons and holes. A UV-3600 ultraviolet–visible spectrophotometer was used to analyze optical absorption (DRS, Shimadzu Group Company, Kyoto, Japan). An Agilent Cary630 Fourier transform infrared spectrometer (FTIR, Shanghai Weiyi Biotechnology Co. Ltd., Shanghai, China) was used to analyze the bonding condition. The electrochemical impedance spectroscopy (EIS) was carried out using an electrochemical workstation (Donghua Test Technology Co. Ltd., Taizhou, China) to investigate the catalyst interfacial charge transfer resistance.

### 2.3. Photocatalytic and Electrochemical Experiment

The photocatalytic performance of the samples was evaluated by taking RhB aqueous solution as the target pollutant. First, 0.1 g photocatalyst was dispersed in 100 mL (10 mg/L) RhB solution to form a mixture. After stirring for 30 min, a 250 W xenon lamp (300–800 nm) was turned on as the UV–visible light source. The lamp was placed 7.5 cm above the liquid level. The mixture was taken every 10 min to measure the absorbance (A) at 553 nm. The degradation degree was computed by the formula (A_0_ − A_t_)/A_0_ × 100%.

After the conductive adhesive was evenly coated on the indium tin oxide (ITO) conductive glass, the sample as dispersed by ethanol ultrasonication was added to the conductive glass and dried in a 100 °C oven to prepare the working electrode. A Ag/AgCl electrode was used as reference electrode, a platinum electrode as auxiliary electrode, and 0.1 mol/L Na_2_SO_4_ aqueous solution as electrolyte. The electrochemical impedance spectroscopy of the catalysts was tested via the electrochemical workstation.

## 3. Results and Discussion

### 3.1. Crystal Structure

Figure 1 displays the XRD patterns of pure TiO_2_ and Ag-TiO_2_. The diffraction peaks of pure TiO_2_ at 25.4°, 36.1°, and 48.0° correspond to the (101), (004), and (200) crystal planes of anatase. The diffraction peaks appearing at 27.4°, 54.4°, and 55.0° correspond to the (110), (105), and (211) crystal planes of rutile. A diffraction peak corresponding to the brookite (121) crystal plane appeared at 30.8°, indicating that the three phases of anatase, rutile, and brookite coexisted in pure TiO_2_ [28,29,30]. In addition to the diffraction peaks of anatase, rutile, and brookite, in the patterns of 1% and 2%Ag-TiO_2_, new diffraction peaks appeared at 38.1° and 44.3° corresponding to the (111) and (200) crystal planes of elemental Ag. As the Ag/Ti molar ratio increased, the diffraction peaks in 4%Ag-TiO_2_ and 6%Ag-TiO_2_ at 27.8°, 32.3°, 46.3°, 55.0°, and 57.1° correspond to the (111), (200), (220), (311), and (222) crystal planes of AgCl [31]. When the Ag/Ti atomic percentages were 1% and 2%, Ag@anatase/rutile/brookite composite heterojunctions formed, and when the percentages were 4% and 6%, Ag/AgCl-modified anatase/rutile/brookite composites formed.

The average grain size was calculated by Scherrer’s formula [32] as follows:(1)$$D=\frac{0.89\lambda }{B\cos\theta }$$
where *D* is the average grain size, *B* is the full width at half maximum (fwhm), 2*θ* is the diffraction angle, and *λ* is the X-ray wavelength (0.15418 nm for Cu target).

The mass fractions of anatase, rutile, and brookite were calculated with Equations (2)–(4):(2)$$W_{a}=\frac{k_{a}A_{a}}{k_{a}A_{a}+A_{r}+k_{b}A_{b}}$$
(3)$$W_{r}=\frac{A_{r}}{k_{a}A_{a}+A_{r}+k_{b}A_{b}}$$
(4)$$W_{b}=\frac{k_{b}A_{b}}{k_{a}A_{a}+A_{r}+k_{b}A_{b}}$$
where *k*_a_ (0.886) and *k*_b_ (2.721) are correction coefficients and *A*_a_, *A*_r_, and *A*_b_ represent the relative intensities of the diffraction peaks of anatase (101), rutile (110), and brookite (121) crystal planes, respectively. The phase composition and average grain size of samples are summarized in Table 1. The effect of Ag addition on the phase transition from anatase to rutile is controversial. Both inhibition and promotion have been reported. Some studies have demonstrated that the deposition of Ag and AgCl particles on the surface of TiO_2_ hinders the migration of Ti and O atoms and delays the nucleation and growth of rutile, showing an inhibitory effect [33,34]. Correspondingly, it has also been reported that the addition of Ag promotes the transformation of anatase to rutile. Scholars have held that the appearance of Ag and AgCl causes lattice distortion of TiO_2_, forms surface defects, and weakens the surface tension of grains, thus promoting phase transformation [35,36]. The content of anatase decreased and rutile content increased after Ag addition, indicating that the transformation of anatase to rutile was promoted by Ag modification, in line with [35,36].

### 3.2. Morphology

Figure 2 depicts SEM images of pure TiO_2_ (a) and 4%Ag-TiO_2_ (b). Both samples showed inconformity in particle size. The agglomerate shape was irregular, and the sizes of agglomerations ranged from tens to hundreds of nanometers. Figure 2c–g is the SEM mapping of 4%Ag-TiO_2_. There were four elements, Ti, O, Ag, and Cl, in the sample, which were basically evenly distributed in the matrix.

Figure 3 shows TEM and HRTEM images of pure TiO_2_ (a,c) and 4%Ag-TiO_2_ (b,d). The single-crystal grain sizes of pure TiO_2_ and 4%Ag-TiO_2_ were both around 10–20 nm. The interplanar spacings in Figure 3c, 0.35 nm, 0.32 nm, and 0.29 nm, correspond to the (101) crystal plane of anatase, the (110) crystal plane of rutile, and the (121) crystal plane of brookite [28,30,37]. Consistently with the XRD results, the pure TiO_2_ was a three-phase coexisting structure of anatase, rutile, and brookite. The interplanar spacing marked in Figure 3d was 0.35 nm, which corresponds to the (101) crystal plane of anatase. The interplanar spacings of 0.23 nm and 0.28 nm were indexed to the (111) crystal plane of elemental Ag and the (200) crystal plane of AgCl [38].

### 3.3. Element Composition

The XPS spectra of pure TiO_2_ and 4%Ag-TiO_2_ are shown in Figure 4. The high-resolution spectra of the 2p peak of Ti are shown in Figure 4b. The binding energies of Ti 2p_3/2_ and Ti 2p_1/2_ of pure TiO_2_ were 458.5 eV and 464.2 eV, respectively. Meanwhile, the binding energies of Ti 2p_3/2_ and Ti 2p_1/2_ of 4%Ag-TiO_2_ were 458.4 eV and 464.2 eV, respectively, indicating that the Ti element had +4 valence [39]. Figure 4c shows the O 1s high-resolution spectra. Pure TiO_2_ showed two characteristic peaks at 529.4 eV and 530.0 eV, corresponding to lattice oxygen (O^2−^) and surface hydroxyl (OH^−^), respectively. The peaks corresponding to lattice oxygen (O^2−^) and surface hydroxyl (OH^−^) were at 529.3 eV and 530.6 eV, respectively, in the 4%Ag-TiO_2_ sample [40]. The high-resolution spectrum of Cl 2p is shown in Figure 4d. The binding energies of the Cl 2p_3/2_ and Cl 2p_1/2_ orbitals of 4%Ag-TiO_2_ were 197.7 eV and 199.0 eV, respectively, indicating that the Cl element had −1 valence [39]. The high-resolution spectrum of Ag 3d (Figure 4e) showed two characteristic peaks at 366.7 eV and 372.9 eV, which were between the standard values of Ag^0^ and Ag^+^ binding energy. Binding energies in the middle of Ag^0^ and Ag^+^ standard values may have been caused by the strong interaction between Ag and AgCl [41,42,43,44].

### 3.4. Surface Area

The specific surface area of photocatalyst is closely related to its photocatalytic activity, as larger specific surface area can provide more active sites, improving the photocatalytic properties. Figure 5 shows the N_2_ adsorption–desorption isotherms and pore size distributions of pure TiO_2_ (a) and 4%Ag-TiO_2_ (b). The pore diameters of the two samples were concentrated between 5 and 20 nm, indicating that the Ag/AgCl modification had little effect on the pore diameter. Plenty of studies have shown that the specific surface area decreases after Ag/AgCl modification. Ag and AgCl nanoparticles occupy part of the pores of TiO_2_, reducing pore volume, which leads to a decrease in surface area. The specific surface area of pure TiO_2_ was 165.9 m^2^/g, which was larger that of 4%Ag-TiO_2_ (149.9 m^2^/g). After Ag/AgCl modification, the specific surface area of TiO_2_ decreased slightly. The results were consistent with the reported literature [20,21,43].

### 3.5. FTIR Analysis

Figure 6 shows the FTIR spectra of pure TiO_2_ and 4%Ag-TiO_2_. The broad absorption band at 3435.07 cm^−1^ was attributed to the O-H stretching vibration, and the broad absorption band at 1635.07 cm^−1^ was attributed to O-H bending vibration [17,45]. The position of the stretching vibration peaks of water and the hydroxyl group hardly shifted after modification, indicating that the structure did not obviously change after Ag addition.

### 3.6. Optical Property

Figure 7 shows the PL spectra of samples. The peak intensity of Ag-TiO_2_ was lower than that of pure TiO_2_, indicating that Ag addition was beneficial for inhibiting the recombination of photoinduced electrons and holes. As the Fermi level of Ag particles is lower than the conduction band of TiO_2_, the photogenerated electrons in the TiO_2_ conduction band migrate to the surface of Ag particles, reducing the recombination probability of photoinduced electron-hole pairs [31,43]. Some studies have shown that there is an optimal concentration for noble metal modification. New recombination centers are formed when the concentration is exorbitant, resulting in an increase in PL peak intensity [17]. On the other hand, Yang et al. [12] found that PL peak intensity declined as Au content increased. In this work, the recombination of 6% Ag-TiO_2_ was the lowest, implying that in a certain range, the higher the Ag/Ti molar ratio was, the higher the photogenerated charge separation of the Ag/AgCl-modified anatase/rutile/brookite composite photocatalysts was.

Figure 8 presents the ultraviolet–visible absorption spectra of pure TiO_2_ and Ag-TiO_2_. Numerous studies have shown that the absorption of visible light increases after Ag modification [11,12,18]. In the present study, the spectra of pure TiO_2_ and Ag-TiO_2_ were substantially identical, indicating that Ag or Ag/AgCl modification did not significantly improve the absorption in visible region. It is worth noting that the UV absorption decreased after the addition of Ag and that the higher the Ag/Ti mole ratio was, the greater the decrease was. The decreased absorption in the UV region can be attributed to the fact that excessive Ag particles covered the surface of TiO_2_ and blocked the light absorption [13].

### 3.7. Photocatalytic Activity

Figure 9 presents the degradation degree curves (a) and kinetics curves (b) of RhB by pure TiO_2_ and Ag-TiO_2_. After illumination for 40 min, the degradation degree of RhB by pure TiO_2_ was 62.2%. The degradation degrees of 1%, 2%, 4%, and 6%Ag-TiO_2_ were 75.2%, 73.2%, 79.6%, and 75.4%, respectively. The degradation degrees of modified TiO_2_ were higher than that of pure TiO_2_, as the addition of Ag was beneficial for improving the separation of photogenerated charges. In particular, 4%Ag-TiO_2_ exhibited the highest degradation degree. XRD results showed that a AgCl phase formed when the molar ratio of Ag/Ti reached 4%. AgCl is able to generate Cl^0^ radicals, which favor photocatalytic activity [20,43]. It is worth mentioning that the degradation degree of 6%Ag-TiO_2_ was less than that of 4%Ag-TiO_2_. An excessive Ag/Ti molar ratio may lead to more Ag particles covering the TiO_2_ surface and hinder the absorption of the light source [13], which was confirmed by the ultraviolet–visible absorption spectra.

Per Figure 9b, time t presented a linear relationship with –ln(C/C_0_), suggesting that the photodegradation reaction conformed to a first-order reaction [44]. The first-order reaction rate constant of pure TiO_2_ was 0.024 min^−1^, and those of 1%, 2%, 4%, and 6%Ag-TiO_2_ were 0.035 min^−1^, 0.033 min^−1^, 0.040 min^−1^, and 0.034 min^−1^, respectively.

Figure 10 exhibits the electrochemical impedance spectroscopy of pure TiO_2_ and 4%Ag-TiO_2_. According to the Nyquist theorem, the diameter of 4%Ag-TiO_2_ is smaller than that of TiO_2_, indicating that it has lower charge movement resistance [46,47,48]. Therefore, 4%Ag-TiO_2_ photocatalyst should exhibit higher photogenerated charge separation efficiency, which was consistent with the PL spectra.

It is documented that Ag nanoparticles exhibit an SPR effect under visible light, resulting in the generation of thermal electrons on Ag particles, some of which cross the energy barrier and transfer to TiO_2_ and AgCl conduction bands to participate in the photocatalytic reaction, improving the photocatalytic performance [39,40,43]. However, in the present work, the ultraviolet–visible absorption spectra showed that there was no obvious SPR effect in Ag-TiO_2_. Therefore, based on characterization and photocatalytic experimental results, we propose a photodegradation mechanism of Ag/AgCl-modified anatase/rutile/brookite composite photocatalysts. A schematic diagram of energy band structure and photogenerated charge transfer of 4%Ag-TiO_2_ is shown in Figure 11 [21,43]. When TiO_2_ is excited by light, generating photoinduced charges, as TiO_2_ is composed of three phases, anatase, rutile, and brookite, photogenerated electrons can migrate rapidly at the phase interfaces, which reduces the probability of recombination with holes. because of the existence of Ag particles, the Fermi level of which is lower than that of the TiO_2_ conduction band, the photogenerated electrons in the TiO_2_ conduction band are injected into the surface of Ag particles to further reduce the recombination with holes [44]. Not only that, the surfaces of AgCl particles are enriched with Cl^−^, and the holes in the AgCl valence band can oxidize Cl^−^ to form Cl^0^ radicals, which process is conducive to increasing the photocatalytic efficiency [43]. Free radicals such as ·OH, ·O_2_^−^, and Cl^0^ can decompose RhB into inorganic small molecules, degrading the target pollutants.

## 4. Conclusions

Pure and Ag/AgCl-modified anatase/rutile/brookite three-phase mixed crystal TiO_2_ composites were prepared by a one-step hydrothermal method. The addition of Ag promoted the transformation from anatase to rutile. The addition of Ag did not improve visible light absorption but was conducive to the separation of photogenerated charges; therefore, the photocatalytic activity increased. When the Ag/Ti molar ratio was 4%, the Ag/AgCl@TiO_2_ exhibited the highest photocatalytic activity. The reaction rate constant k of 4%Ag-TiO_2_ was 0.040 min^−1^, which was higher than that of pure TiO_2_ (0.024 min^−1^).

## Figures and Tables

**Figure 1 nanomaterials-12-01599-f001:**
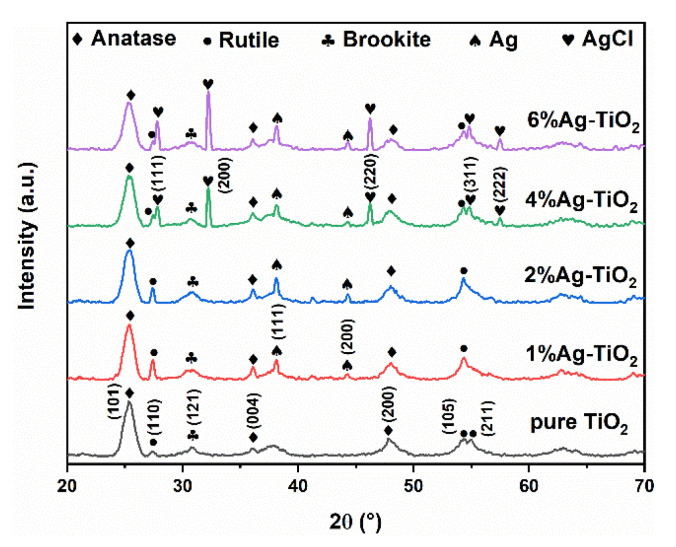
XRD patterns of pure TiO_2_ and Ag-TiO_2_.

**Figure 2 nanomaterials-12-01599-f002:**
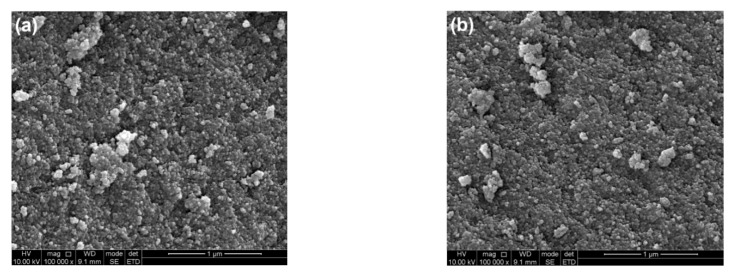
SEM images of (**a**) pure TiO_2_ and (**b**) 4%Ag-TiO_2_ and SEM mapping of (**c**–**g**) 4%Ag-TiO_2_.

**Figure 3 nanomaterials-12-01599-f003:**
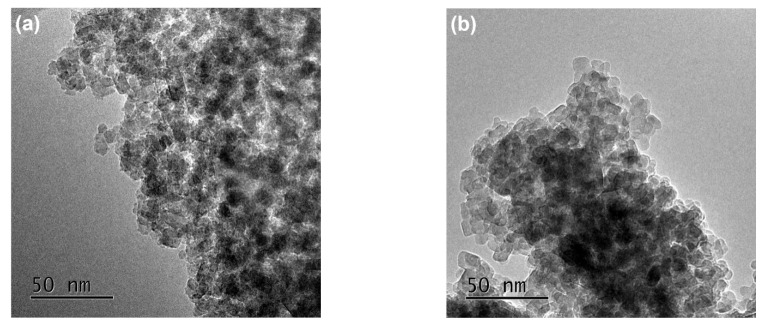
TEM and HRTEM images of (**a**,**c**) pure TiO_2_ and (**b**,**d**) 4%Ag-TiO_2_.

**Figure 4 nanomaterials-12-01599-f004:**
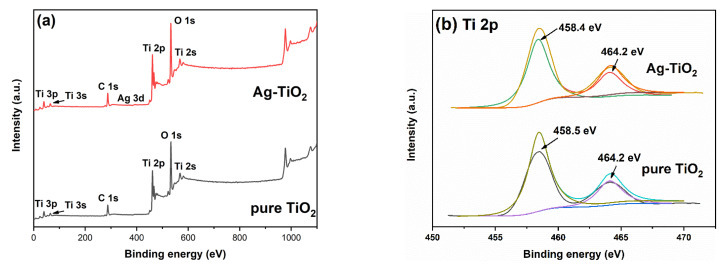
XPS spectra of pure TiO_2_ and 4%Ag-TiO_2_: total spectra (**a**), Ti 2p (**b**), O 1s (**c**), Cl 2p (**d**) and Ag 3d (**e**).

**Figure 5 nanomaterials-12-01599-f005:**
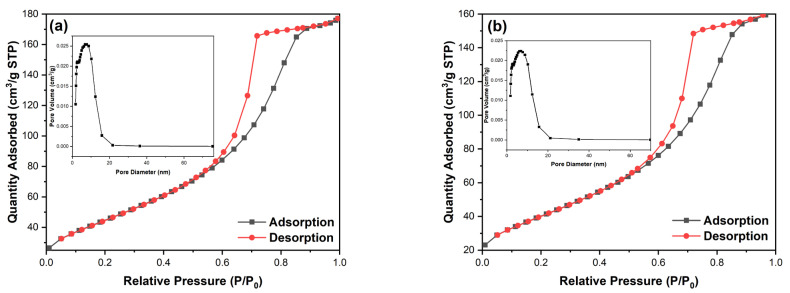
Nitrogen adsorption–desorption isotherms and pore size distribution curves of (**a**) pure TiO_2_ and (**b**) 4%Ag-TiO_2_.

**Figure 6 nanomaterials-12-01599-f006:**
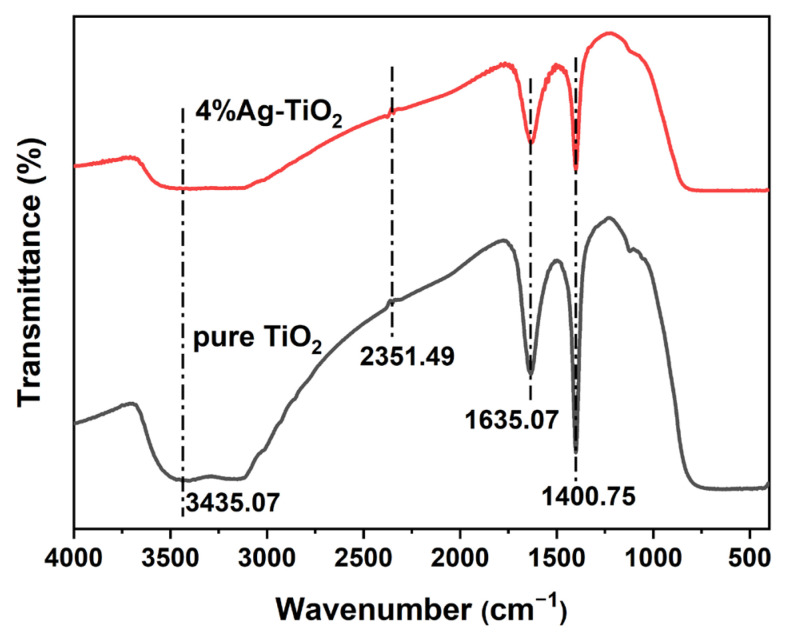
FTIR spectra of pure TiO_2_ and 4%Ag-TiO_2_.

**Figure 7 nanomaterials-12-01599-f007:**
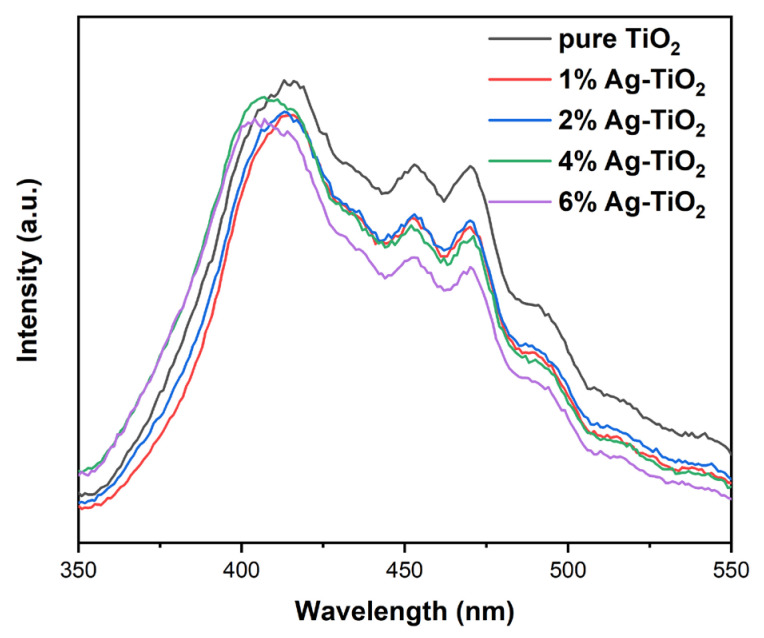
Photoluminescence (PL) spectra of pure TiO_2_ and Ag-TiO_2_.

**Figure 8 nanomaterials-12-01599-f008:**
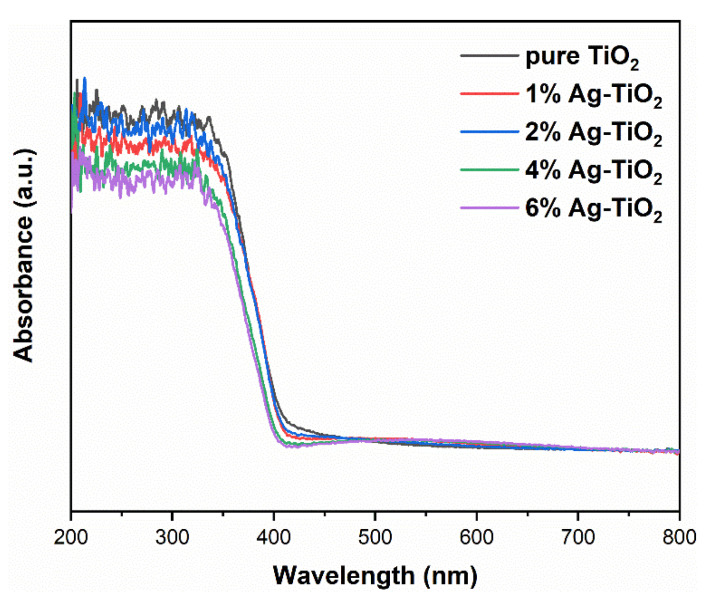
DRS spectra of pure TiO_2_ and Ag-TiO_2_.

**Figure 9 nanomaterials-12-01599-f009:**
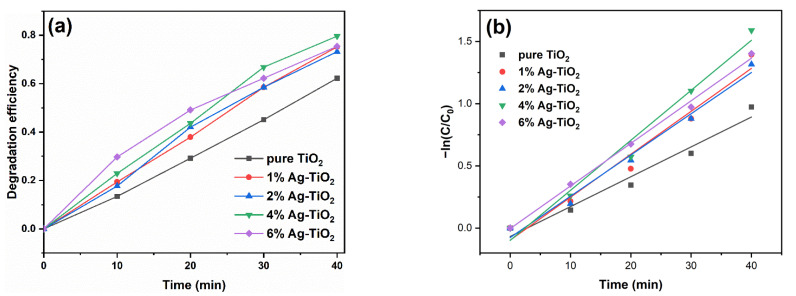
(**a**) Degradation degree curves and (**b**) kinetics curves of pure TiO_2_ and Ag-TiO_2_.

**Figure 10 nanomaterials-12-01599-f010:**
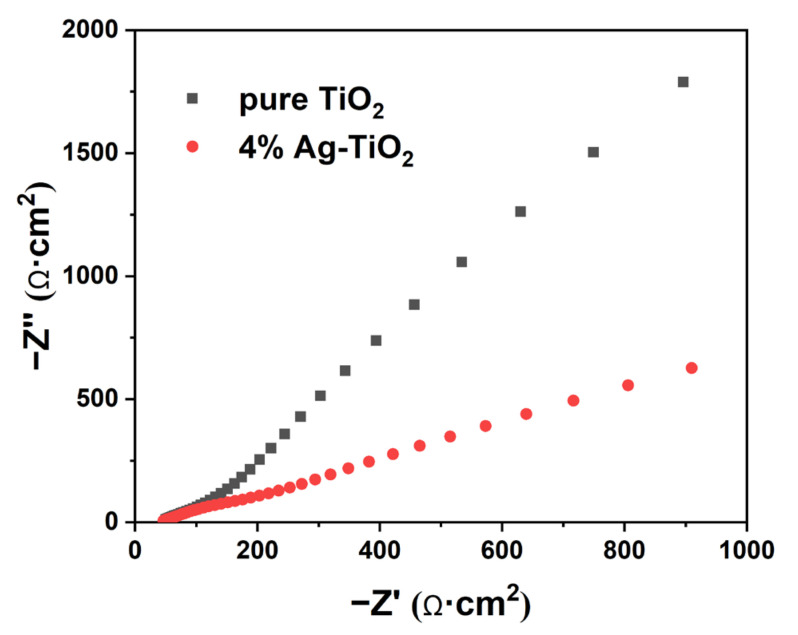
EIS Nyquist plots of pure TiO_2_ and 4%Ag-TiO_2_.

**Figure 11 nanomaterials-12-01599-f011:**
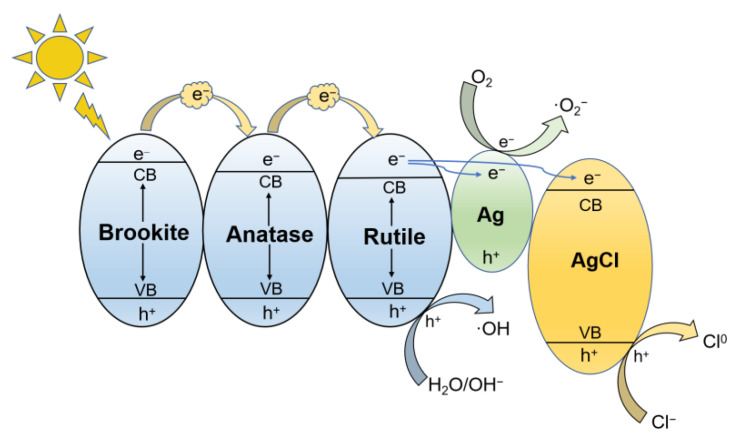
The energy band structure and the diagram of photogenerated charge transfer of Ag/AgCl@TiO_2_.

**Table 1 nanomaterials-12-01599-t001:** Phase composition and average crystallite size of pure TiO_2_ and Ag-TiO_2_.

Samples	Phase Composition (%)/Crystallite Size (nm)		
	Anatase/Crystallite Size	Rutile/Crystallite Size	Brookite/Crystallite Size
pure TiO_2_	70.0%/9.8 ± 0.4	9.0%/46.2 ± 1.1	21.0%/11.4 ± 1.8
1%Ag-TiO_2_	53.1%/10.2 ± 0.1	25.4%/25.7 ± 6.4	21.5%/9.9 ± 1.5
2%Ag-TiO_2_	46.3%/9.6 ± 0.7	22.6%/29.5 ± 6.5	31.1%/12.4 ± 1.2
4%Ag-TiO_2_	50.0%/10.1 ± 0.2	24.3%/29.3 ± 6.1	25.7%/13.3 ± 0.7
6%Ag-TiO_2_	52.2%/9.7 ± 0.3	28.0%/35.3 ± 3.8	19.8%/14.1 ± 2.9

## Data Availability

Data are contained within the article.

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
