# Peer review of "Three-Phase Mixed Titania Powder Modified by Silver and Silver Chloride with Enhanced Photocatalytic Activity under UV–Visible Light"

_nanomaterials, 2022, doi:10.3390/nano12091599_

Round 1

Reviewer 1 Report

The authors describe a one-step hydrothermal method of Ag/AgCl modified titania powders with three phases, their characterization and their photocatalytic activity.

I am sorry to state that the manuscript has not the novelty and scientific soundness to be published in Nanomaterials.

Some suggestion to improve the manuscript in order to be published in other journals:

  • an extensive editing of English language and style is required.
  • the presence of SPR of Ag should be included in abstract.
  • the Experiment section is incomplete. Material preparation (one of the most important issues in manuscript) should be far more detailed. 
  • FTIR technique could be useful to enlarge Characterization section.
  • Photocatalytic experiment section is also incomplete. You do not describe reaction system, nor volumes taken in each sample, etc.
  • The changes in phase composition when Ag concentration varies are not properly explained.
  • The reference in text of Figure 2 is not correct.
  • The changes in BET areas after Ag/AgCl modification are not explained.
  • The decrease in absorbance with increasing Ag concentration is really strange. 
  • The results in photocatalytic degradation are not fully explained (the results for all Ag modified titania powders are almost the same, the absence of SPR effect of Ag is not supported, etc.).
  • Apart from photoluminescence you could apply electrochemical impedance spectroscopy in order to a deeper understanding of the separation of photogenerated charges in each catalyst.

Author Response

Responses to  the reviewer's comments are attached.

Reviewer 2 Report

This is a fairly solid work, which should undoubtedly be recommended for publication, but after clarifying some incomprehensible points.

  1. Line 23. It is important to note here that many different impurities in TiO2 have been tested and they show different effects. Note that many such studies have been published in MDPI journals:

Serga, V.; Burve, R.; Krumina, A.; Romanova, M.; Kotomin, E.A.; Popov, A.I. Extraction–Pyrolytic Method for TiO2 Polymorphs Production. Crystals 202111, 431. https://doi.org/10.3390/cryst11040431

Sun, X.; Yan, X.; Su, H.; Sun, L.; Zhao, L.; Shi, J.; Wang, Z.; Niu, J.; Qian, H.; Duan, E. Non-Stacked γ-Fe2O3/C@TiO2 Double-Layer Hollow Nanoparticles for Enhanced Photocatalytic Applications under Visible Light. Nanomaterials 202212, 201. https://doi.org/10.3390/nano12020201

and many others.

  1. Table 1. it is necessary to indicate the measurement error, especially for the crystallite size.
  2. Whether the temporal evolution of SEM pictures was observed (paragraph 3.2) ?
  3. Paragraph 3.5. Luminescence excitation wavelength not specified. The data in Figure 6 is not deciphered and there is no comparison with other experiments.

Author Response

(The authors gave the same response as above.)

Round 2

Reviewer 1 Report

Thank you very much for your efforts to improve the manuscript. I think now it has the quelite to be published in Nanomaterials,

Reviewer 2 Report

The authors have successfully and fully responded to all comments / questions, so that the article can be recommended for publication